# Review of Seawater Fiber Optic Salinity Sensors Based on the Refractive Index Detection Principle

**DOI:** 10.3390/s23042187

**Published:** 2023-02-15

**Authors:** Gaochao Li, Yongjie Wang, Ancun Shi, Yuanhui Liu, Fang Li

**Affiliations:** 1State Key Laboratory of Transducer Technology, Institute of Semiconductors, Chinese Academy of Sciences, Beijing 100083, China; 2College of Materials Science and Opto-Electronic Technology, University of Chinese Academy of Sciences, Beijing 100049, China

**Keywords:** optical fiber sensor, salinity, refractive index, ocean observation, absolute salinity

## Abstract

This paper presents a systematic review of the research available on salinity optic fiber sensors (OFSs) for seawater based on the refractive index (RI) measurement principle for the actual measurement demand of seawater salinity in marine environmental monitoring, the definition of seawater salinity and the correspondence between the seawater RI and salinity. To further investigate the progress of in situ measurements of absolute salinity by OFSs, the sensing mechanisms, research progress and measurement performance indices of various existing fiber optic salinity sensors are summarized. According to the Thermodynamic Equation of Seawater-2010 (TEOS-10), absolute salinity is recommended for sensor calibration and measurement. Comprehensive domestic and international research progress shows that fiber-optic RI sensors are ideal for real-time, in situ measurement of the absolute salinity of seawater and have excellent potential for application in long-term in situ measurements in the deep ocean. Finally, based on marine environmental monitoring applications, a development plan and the technical requirements of salinity OFSs are proposed to provide references for researchers engaged in related industries.

## 1. Introduction

Seawater salinity is one of the most important parameters used for studying the nature of seawater, and its distribution changes affect and constrain the distribution and changes of hydrological [1], chemical [2], biological [3] and other elements. Studies of seawater movement [4], delineation of water masses [5], and the determination of sound velocity [6] are closely related to salinity distribution and change patterns, so accurate in situ measurements of seawater salinity are necessary. At the same time, as marine environmental monitoring is gradually extending to the deep sea, the conditions for in situ seawater salinity measurements are becoming more demanding, so traditional seawater salinity measurement equipment and methods for deep sea observation suffer from challenges, such as strict requirements for resolution, accuracy, and long-term stability.

The basic definition of seawater salinity is the amount of dissolved salt in 1 kg of brine in g/kg or ‰. The composition of seawater salinity is complex and diverse, and its distribution varies in different seas. In terms of definition, it has undergone a long standardization process using the international practical salt standard PSS78 [7] and seawater thermodynamic equation 1980 (EOS-80) [8]. The practical salinity defined by the practical salinity scale uses the principle of constancy of seawater components (Marcet’s principle). This principle can be applied to the overall analysis of seawater but is clearly not applicable to in situ measurement needs. The practical salinity unit (PSU) is the standard for reporting practical salinity in oceanography and is unitless, generally being expressed in ‰. Absolute salinity can be described by the new seawater thermodynamic equation, and TEOS-10 [9] defines absolute salinity as “the mass fraction of a dissolved nonaqueous material in a saltwater sample under particular temperature and pressure circumstances,” which can more realistically and precisely represent seawater attributes and allow direct traceability. The unit of absolute salinity is the same as that of basic salinity, which is g/kg or ‰. In this paper, ‰ will be used as the unit of salinity. Absolute salinity and practical salinity can be interconverted under known longitude, dimension and pressure, and their difference is indicated as the salinity correction (*δS_A_*). As shown in Figure 1, the maximum difference of 0.026 g/kg in the density profile of seawater was calculated globally using the absolute salinity of EOS-80 and TEOS-10, which seriously affects the conclusions defined by the physical ocean study [10].

The traditional methods of seawater salinity measurement include evaporation weighing [11], chlorinity titration [12] and conductivity [13]. Neither evaporation weighing nor chlorine titration can be used for in situ measurements, and these methods do not meet the needs of current applications. The conductivity method is the most widely used in current applications. However, it ignores nonconductive neutral substances, requires high temperature compensation of the sensor and has problems with long-term zero drift. The conductivity method has failed to meet the current needs of in situ measurements of seawater salinity.

New measurement methods are currently needed to enable in situ measurements of seawater salinity. For example, vibrating tube dosimeters (VTDs) [14] and pycnometry [15] are used to measure seawater density but are difficult to apply to seawater salinity in situ measurements. The speed of sound method [16] can be applied to seawater salinity in situ measurements. However, the uncertainty of measurements by conducted this method is close to 200 ppm at atmospheric pressure and up to 300 ppm at high pressure [17]. In the salinity range of 0–50‰, the accuracy of the three-stage standard conversion of conductivity for salinity measurements is 71.42 ppm, and the accuracy of the speed of sound method does not meet the requirements of seawater salinity measurements. The RI method of salinity measurement has the sensitivity of the whole media component. The measured RI results have a strong correspondence with the absolute salinity. Philippe Grosso [18] described the case of an optical salinity sensor for the direct measurement of the RI of seawater, detailing the dependence of the measurement on environmental parameters (especially temperature and pressure) and demonstrating that it may be advantageous to measure the RI directly instead of the conductivity, thus achieving a more direct pathway to absolute salinity. The RI method is the closest field application to the absolute salinity in terms of measurement accuracy and the need for in situ measurements in the deep sea. Overall, by combining various measurement methods, the RI method is currently the most promising means of measuring absolute salinity. Recently, the development of OFS technology [19] has gradually matured, and its mechanism lies in the use of the RI method to measure salinity, which is consistent with the measurement method of absolute salinity. Therefore, the in situ measurement of seawater absolute salinity based on the OFS scheme has a very good prospects, which will replace the conductivity measurement of practical salinity and is likely to become the mainstream method for the in situ measurement of seawater absolute salinity in the future. However, the current salinity OFS is still under development and there is currently no mature in situ measurement scheme. In this paper, the research progress of salinity OFSs based on the principle of RI measurement introduced in recent years is reviewed, which inspires some ideas for further research on the in situ measurement of seawater salinity. This review consists of four sections. Section 1 illustrates the definition of salinity and the advantages of the RI method for measuring absolute salinity. Section 2 describes the conversion relationship between the RI and seawater salinity. In Section 3, salinity OFSs with different mechanisms are introduced. Section 4 summarizes the current indicators of salinity OFSs and comprehensively compares and analyzes the characteristics of various types of sensors. Finally, the development trend and technical needs of salinity OFSs are discussed to provide references for researchers engaged in related industries.

## 2. Principle

The correspondence between seawater salinity and RI has been studied more extensively in the last century. In 1976, R.W. Austin and G. Halikas [20] summarized the correspondence tables of RI (n), salinity (S), pressure (P), temperature (T), and wavelength (λ) and used first-order gradient interpolation to supplement the increment table. Their main work was to establish tabular data for the five parameters of n, S, P, T and λ, which is very meaningful work. However, the process of looking up parameters in tables is more cumbersome in practical applications. The wavelength applicable range is only 400~700 nm, that for salinity is 0~43‰ (practical salinity), that for temperature is 0~30 °C, and that for pressure is 0~1100 kg/cm^2^.

In 1990, R.C. Millard and G. Seaver [21] developed a RI salinity correspondence algorithm suitable for pure water and seawater using four sets of experimental data with different accuracies. Its wavelength range is 500~700 nm, the temperature range is 0~30 °C, the salinity range is 0~40 (practical salinity), and the pressure range is 0~11,000 db. The the Seaver and Millard algorithm is as follows:(1)$$n(T,P,S,\lambda )=n_{I}(T,\lambda )+n_{\mathrm{II}}(T,\lambda ,S)+n_{\mathrm{III}}(P,T,\lambda )+n_{\mathrm{IV}}(S,P,T)$$

The accuracy of the RI algorithm ranges from 0.4 ppm for pure water at atmospheric pressure to 80 ppm at high pressure but retains the accuracy of each original dataset. The algorithm introduces pressure parameters, which can be used to convert seawater salinity and RI at nonatmospheric pressure, which is a major improvement in the correspondence between salinity RI at that time, only improving the conversion accuracy but also reducing the cumbersome degree in actual use.

In 1995, X. Quan and E.S. Fry [22] discussed two preexisting empirical formulas for the RI of seawater, and Figure 2a–c below show that the error range of different fitting methods is gradually narrowing; however, the algorithm lacks pressure parameters in the quantitative relationship and cannot be applied to the in situ measurement of different pressure environments in the ocean.

NKE, a French company, first released an in situ measurement product for determining the absolute salinity of seawater, NOSS [23], in 2009, with an in situ RI measurement accuracy of 1.1 × 10^−6^ and an absolute salinity measurement resolution of 0.005‰. The product has been tested in deep-sea comparisons. Its RI and salinity conversion method applies the Seaver and Millard algorithm [21]. NOSS uses the operating principle of prismatic spectroscopy and its schematic and physical properties, as shown in Figure 3. The NOSS sensor uses a conventional optical method to measure the RI of seawater, which fills the notch of the prism during operation. The light source used for the sensor system is a diode laser. The light reaches the reflector made by the Au deposition method at a, reflects and is influenced by the RI of seawater at b to change the position of the light and is then reflected and received by the PSD at c to obtain the change in the position of the light and thus determine for the RI of seawater.

## 3. Salinity OFSs

Based on the above correspondence between salinity and the RI, the salinity of seawater can be measured indirectly by measuring the RI. To improve the salinity measurement performance of fiber optic sensors as well as their applications, many methods have been proposed in recent years. This paper will review in detail the structure, basic principles, advantages, and disadvantages of different methods.

### 3.1. Optical Fiber SPR

Fiber surface plasmon resonance (SPR) refers to the plasmonic excitation effect that takes place between the swift wave on the fiber surface and the metal film that has been plated on the fiber surface. Typically, less than 100 nm of the metal film, such as gold, silver, alumina and titanium dioxide, is plated on the surface of the swift field transmission fiber to produce this effect. The wave vector matching criterion, which is connected to the surface, must be satisfied for the metal surface plasmon wave and the fiber fast field wave vector to generate SPR. This is related to the effective RI of the metal on the surface, which is controlled by the RI of the surroundings in contact with the metal sheet [24].

Various structures and materials are used to improve the performance of SPR sensors. In 2019, Zhao [25] et al. reported an optical fiber SPR sensor for the simultaneous measurement of seawater salinity and temperature (Figure 4a). The sensitive detection zone for SPR was a nanoscale gold sheet. The sensitive layer was partially covered with polydimethylsiloxane (PDMS) to generate double SPR peaks (Figure 4a). To create a full detection system, the SPR effect model of the fiber structure was updated. The salinity precision was less than or equal to 0.3 on average. In 2020, Siyu E. et al. [26] proposed an HCF-based two-channel SPR sensor for the simultaneous measurement of seawater salinity and temperature. One SPR channel excited on the outer surface of the hollow core fiber (HCF) with an Au membrane coating is used to measure seawater salinity, while the other SPR channel excited on the inner surface of the HCF with a Ag membrane coating and PDMS permeation is used to simultaneously measure seawater temperature (Figure 4b). This approach shows the feasibility of simultaneously measuring multiple parameters with high sensitivity, high stability, small size, and high mechanical strength.

Due to its good sensing performance and unique air hole structure, a photonic crystal fiber (PCF) can be filled with a variety of temperature-sensitive materials to achieve the dual-parameter sensing of salinity and temperature [27]. Even simultaneous detection of salinity, temperature, and pressure in seawater is achieved. For example, in 2019, Yong Zhao [28] used two sensitive membranes (PDMS, SU-8) coated on different parts outside the Au membrane, thus forming three different sensitive regions, and then three different SPR resonance depressions appeared upon optimizing the parameters of the structure. The maximum sensitivities of this method for salinity, temperature and pressure measurements were 0.560 nm/‰, 1.802 nm/°C, and 2.838 nm/MPa, respectively. Based on other structures, in 2021, Yang [29] et al. proposed a surface plasmon resonance (SPR) sensor based on an exposed core microstructured optical fiber (EC-MOF) (Figure 4c) for temperature self-compensated salinity detection. In particular, they used a finite element method to investigate and optimize the sensing performance for 1.33–1.39 with maximum wavelength sensitivities of 2000 nm/RIU and 3000 nm/RIU, respectively. The temperature self-compensated salinity sensing capability was demonstrated with high sensitivities of 4.45 nm/% and −0.12%/°C. This is the first time that a fiber optic SPR sensor with a single sensing channel and a single demodulation method has been temperature self-compensated. PCF sensors based on SPR technology [30,31] have been widely used recently. PCF has more structural adaptability as a sensor head, K. C. Ramya [32] used dual-core PCF scheme and explored the sensing mechanism based on mode coupling between two cores (liquid core and fiber center core). Asif Zuhayer [33] then added the dual-core D-Formed PCF scheme with a sensitivity of up to 18,800 nm/RIU.

In addition to the abovementioned schemes that have been applied to salinity measurements, some schemes have been studied only for RI measurements. Based on the correspondence mentioned above, relevant research advances can be drawn upon. In 2020, Wen Li [34] performed a theoretical analysis of an oxide (MgO, TeO_2_, and TiO_2_) fiber optic SPR sensor on an Au film (Figure 4d). The simulation results showed that by adding oxides to the pristine Au film, it was possible to simultaneously improve the sensitivity and tune the resonant wavelength to a longer wavelength. The sensitivity was greater than 9000 nm/RIU, and the resonant wavelength covered the communication C+L band (1530 nm–1625 nm). Li [35] proposed a reflective seawater salinity and temperature sensor based on the SPR effect in a multimode single-mode (MS) fiber structure. Similarly, PDMS was used as a temperature-sensing material, and its sensitivity was demonstrated to be 0.31 nm/‰ and −2.02 nm/°C. In terms of results, the proposed sensor optimized the cross-sensitivity between salinity and temperature.

Overall, the fiber optic SPR sensor has the advantage of high sensitivity. However, long-term seawater in situ measurements are currently not possible due to the poor corrosion resistance of the Au film.

Table 1 provides a summary of optical fiber SPR sensors, in which the application, year, technique, size and other relevant parameters are listed in categories. In particular, the parameters are given in nm/% or nm/RIU depending on whether the sensor tested for salinity or RI, and some of the salinity units are also available in other cases, as noted. Regarding the size of the sensor, since the size of most fiber optic sensors depends on the package size, the bare fiber diameter is approximately 125 μm, so the size stated here is the package size. For the other parameters, most of the sensors are in the laboratory stage, the temperature is room temperature, and special cases are indicated. The situation is similar in Table 2, Table 3, Table 4, Table 5, Table 6, Table 7 and Table 8.

**Figure 4 sensors-23-02187-f004:**
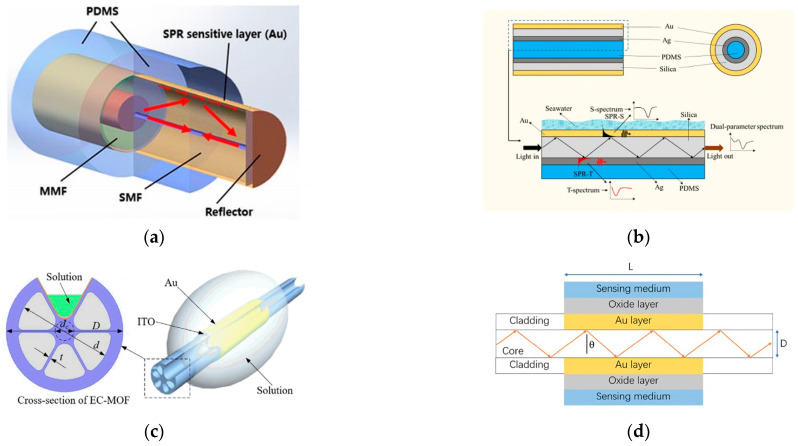
(**a**) Structure of the composite reflection probe [25]. (**b**) Structure and sensing mechanism of the two-channel SPR sensor based on HCF [26]. (**c**) Schematic of the EC-MOF sensor [29]. (**d**) Schematic diagram of the SPR-based optical fiber sensor [34].

### 3.2. Optical Grating

A fiber grating is a passive filter device that forms a diffraction grating by modulating the RI of an optical fiber core axially and periodically by certain methods. Depending on the length of the written grating, fiber gratings can be divided into short-period fiber gratings (FBGs) and long-period fiber gratings (LPGs).

#### 3.2.1. FBG

An FBG is a wavelength-modulated device, and when the external environment (such as salinity and temperature) changes, its transmission characteristic spectrum will change. By analyzing the characteristic wavelength, the sensing function of seawater salinity and temperature can be realized.

Etching is the most commonly used process in FBGs, and in 2016, Ming Chang Shih [36] proposed a method for FBGs to measure the RI of liquids by depleting the cladding of the FBG through etching to enhance the RI measurement sensitivity. In 2017, Jean [37] used the same etching scheme in combination with a Vis-FBG using the peak located at 673.07 nm, which was demonstrated to be less sensitive and less resolved than that of an FBG of a similar diameter operating in the IR range. In 2018, Aliya Bekmurzayeva [38] proposed a wet-etched fiber Bragg grating (EFBG) method to fabricate low-cost low-precision RI sensors with resolutions up to 10^−4^ RIU. In 2020, Yadvendra Singh [39] observed particles of reduced graphene oxide (rGO) attached to the surface of the EFBG, resulting in enhanced refractive sensing ability on the sensor surface. The following year, Yadvendra Singh [40] continued to use the EFBG scheme for additional salinity measurements, refining the EFBG salinity sensor.

In seawater salinity measurements, temperature and salinity parameters are cross-sensitive, so dual-parameter measurement schemes for temperature and salt have been widely proposed. In FBG-based sensors, coating schemes can be used to achieve dual-parameter temperature and salt measurements. In 2008, Men et al. [41] used a dual fiber grating cascade sensor design scheme to achieve simultaneous sensing of seawater temperature and salinity, where the salinity sensitivity was approximately 1.6 pm/‰ and the temperature sensitivity was approximately 16.5 pm/°C. The sensor structure is shown in Figure 5c, where one grating is coated with polyimide to form the salinity sensing zone and the other grating is coated with an acrylic film that is only sensitive to temperature. This scheme was developed in the single-parameter EFBG and hydrogel technique [42,43]. In 2017, Dong Luo et al. combined this grating coating structure with the etching scheme described above to further enhance its salinity and temperature measurement sensitivity [44]. The advantage of the coating scheme is that it can be adapted to different parameters to suit different measurement requirements and environments.

With the innovation of different optical fiber structures, these structures are also increasingly used in FBG salinity sensors. In 2019, Xiaohe Li [45] presented a theoretical study of a new RI sensor based on FBG in NBF. The spectral response of the FBG in the NBF to filled RI in different aperture sizes and RI ranges was investigated by means of the finite element method and the TMM method. Their results show that the aperture diameter has a considerable influence on the sensitivity of the Bragg wavelength to filled RI and the detectable resolution of RI, and increasing the aperture diameter can improve the wavelength sensitivity. In the same year, Peixuan Tian et al. [46] proposed a multiparameter sensor based on the HSCF-FBG scheme with a RI sensitivity of 8.9 nm/RIU, and its overall structure is similar to that of a conventional single-mode fiber FBG. Notably, the bending characteristics of the HSCF-FBG are highly dependent on the bending direction, with a maximum bending sensitivity of 84.11 pm/m. Therefore, the proposed HSCF-FBG can also be used as a bending sensor with directional recognition. The development of this direction is beneficial for the integration of seawater salinity sensors with other parameter sensors. Guofeng Sang [47] developed a compact multifunctional fiber optic sensor based on single-mode fiber-photonic crystal that maintains a fiber Bragg grating (SPNPS-PMFBG) (Figure 5d). By monitoring the wavelength and output power of the two PMFBG resonance peaks, simultaneous measurements of salinity, temperature and strain are possible. The two response times are 3 s at 0–7 wt% salinity, and the sensitivity is 0.58 dB/wt%.

#### 3.2.2. LPG

The transmittance of LPG spectra is modulated by the RI of the surrounding environment, making it suitable for monitoring environmental changes and biological applications [48], i.e., direct seawater salinity sensing via the RI.

In 2014, G. E. Silva et al. [49] reported the development of an LPG fiber optic sensor for RI measurements, namely, a superimposed long period grating (SLPG) (Figure 5e) in standard optical fibers with an internal connection that uses electric arc technology, which can better reduce the cross-sensitivity of temperature and RI. In 2016, Chao Du et al. [50] reported a new device for measuring the internal RI using PCF long-period gratings. A high sensitivity value of 2343 nm/RIU and a high measurement resolution of 8.5 × 10^−6^ RIU were obtained when the RI was measured using a resonant tilting paraflap with a 2.7 cm long period grating and a 180 μm grating period. The sensor is compact, and other features may be important for application in some biological and chemical fields. In 2018, Zhong-Ming Zheng et al. [51] used a small LPG with a period of 40 μm prepared in a single-mode fiber by direct writing with a femtosecond laser. In the broadband spectral range, a series of higher-order Bragg resonance peaks of the Bragg grating and the attenuation band of the LPG were observed simultaneously, which showed different responses to the surrounding RI, temperature and axial strain. Additionally, a sensitivity matrix was provided to correct for temperature-induced errors in RI and strain measurements for a two-parameter fiber optic sensor at high temperatures. In the same year, An Jia et al. [52] designed and fabricated a high-resolution long-period fiber grating RI sensor. A cascaded long-period fiber grating scheme combined with a high-precision demodulator was used to improve the measurement resolution; a higher subcladding mode and an etched fiber cladding were used to improve the long-period fiber grating RI sensitivity. In 2019, Yani Zhang et al. [53] also used femtosecond laser pulse processing based on an 800 nm wavelength and 100 fs pulse width, allowing simultaneous measurement of temperature, RI, and strain changes (Figure 5f).

Overall, the FBG scheme is not as sensitive as the base, but the grating structure is more stable and can be applied to the marine in situ measurement environment.

**Figure 5 sensors-23-02187-f005:**
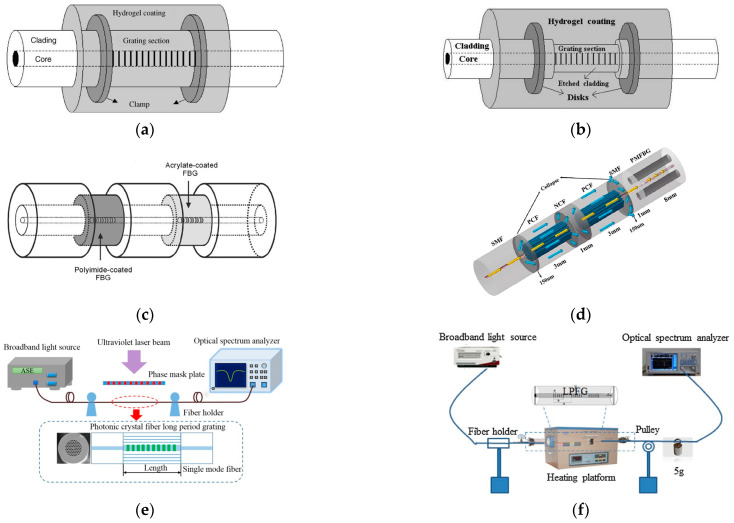
(**a**) Structure of the hydrogel-coated FBG sensor [42]. (**b**) Structure of the hydrogel-coated etched FBG sensor [43]. (**c**) Illustration of the multiplexed FBG sensor system [41]. (**d**) Diagram of the fusing process at different stages [47]. (**e**) Schematic of the measurement system and its sensing structure [49]. (**f**) Schematic of the measurement system of sensing sensitivity [53].

**Table 2 sensors-23-02187-t002:** Summary of optical gratings.

Application	Year	Reference	Technique	Parameters	Others	Size (Diameter)
Salinity sensor	2002	[42]	FBG and hydrogels	sensitivity: - range: 0.1 to 0.7 mol/L	wavelength: 1556 nm temperature: -	-
Salinity sensor	2003	[43]	EFBG and hydrogels	sensitivity: 10.4pm/‰ range: 0.1 to 0.7 mol/L	wavelength: 1550 nm temperature: -	-
RI sensor	2017	[37]	Etched MMF Vis-FBG	sensitivity: 15.71 nm/RIU range: 1.3328 to 1.4607	wavelength: 450–2400 nm temperature: 20 °C	7.00 ± 0.14 μm
Salinity and Temperature sensor	2008	[41]	Polymer-coated and acrylate-coated FBG	sensitivity: 125.92 nm/RIU range: 0 to 5.70 mol/L	wavelength: 1550 nm temperature: 20–90 °C	-
Salinity and Temperature sensor	2017	[44]	Polymer-coated EFBG	sensitivity: 0.0165 nm/M range: 0 to 5.70 mol/L	wavelength: 1535–1555 nm temperature: 20 °C	20 μm
RI sensor	2019	[45]	NBF-FBG	sensitivity: 53.6923 nm/RIU range: 1.0 to 1.48	wavelength: 1550–1570 nm temperature: -	-
RI and Temperature sensor	2019	[46]	FBG and HSCF	sensitivity: 8.9 nm/RIU range: 1.41 to 1.44	wavelength: 1530–1560 nm temperature: -	132 μm
Salinity, Temperature, and strain sensor	2022	[47]	SPNPS-PMFBG	sensitivity: 0.58 dB/wt% range: 0 to 7 wt%	wavelength: 1540–1580 nm temperature: −20–180 °C	-
RI and Temperature sensor	2010	[49]	Superimposed long-period gratings (SLPGs)	-	wavelength: 1551–1559 nm temperature: -	-
RI sensor	2016	[50]	LPG and PCF	sensitivity: 2343 nm/RIU range: 1.3333 to 1.3792	wavelength: 1400–1600 nm temperature: -	-
RI and Temperature sensor	2018	[51]	SP-LPGs	sensitivity: 1178.6 nm/RIU range: 1.4050 to 1.412	wavelength: 1500–1550 nm temperature: 24 °C	-
RI and Temperature sensor	2018	[52]	LPG and PAA	sensitivity: 1178.6 nm/RIU resolution: 10^−6^ RIU	wavelength: 1515–1555 nm temperature: 12–30 °C	119.04–127.37 μm
Salinity, Temperature, and strain sensor	2019	[53]	Femtosecond laser and SMF-28 LPG	sensitivity: −582.5 nm/RIU range: 1.342 to 1.380	wavelength: 1250–1550 nm temperature: 20–800 °C	-
RI sensor	2016	[36]	Cladding depleted FBG	sensitivity: −582.5 nm/RIU range: 1.342 to 1.380	wavelength: 980 nm temperature: -	16.0–35.0 μm
RI sensor	2018	[38]	Etched Fiber Bragg Grating (EFBG)	resolution: ~10^−4^ RIU range: 1.4	wavelength: 1566.8 nm temperature: -	-
Salinity sensor	2020	[39]	Graphene Oxide (rGO) coated EFBG	sensitivity: 3.99 nm/RIU range: 25%	wavelength: 1566.8 nm temperature: 20 °C	
Salinity sensor	2021	[40]	EFBG	sensitivity: 1.825 nm/RIU range: 25%	wavelength: 1546 nm temperature: 25 °C	50 μm

### 3.3. Interferometer

A salinity OFS based on an interferometer that is highly sensitive to the external RI has been proposed. When measuring solution salinity, the interferometric fiber optic sensor is not only relatively simple and sensitive to fabrication but can also be used for long-term measurements.

In an interferometer, there are two beams of light with the same frequency, constant phase difference and the same transmission direction, and the effective phase difference between the two beams will change after a change in the salinity of the solution, which in turn will lead to a shift in the interference spectrum. According to the type of structure, interferometers used for salinity sensing can be classified as Mach–Zehnder interferometers (MZIs), Fabry–Perot interferometers (FPIs) and Sagnac interferometers (SIs).

#### 3.3.1. F-P Interferometer

For FPI structures, the sensing mechanism is mainly based on the analysis of the reflected power from the fiber tip, which varies with the liquid RI. As early as 2008, Z. L. Ran et al. [54] proposed a F-P fiber tip sensor for high-resolution RI measurements. In Figure 6a, “1”, “2” and “3” denote three cavities, and the refractive indices of the fiber and liquid are denoted as n0 and n′, respectively. n′ can be obtained from the cavity length calculation. In 2010, Hae Young Choi et al. [55] proposed a new dual-parameter sensor for salinity and temperature based on a dual-cavity FPI. The sensing probe consists of two cascaded F-P cavities, which has the advantage that the fabricated sensor head is very compact, and the total length of the sensing section is less than 600 μm. Since the reflection spectrum of the composite FP structure is given by the superposition of the spectra of each cavity, both temperature and salinity can be measured independently (Figure 6b). The RI is 16 dB/RIU in the RI range of 1.33–1.45.

Advances in fiber-optic etching and fusion techniques have also led to new solutions, and the following year, Linh Viet Nguyen [56] reported an FPI fiber-optic sensor for the simultaneous measurement of temperature and water salinity. The scheme mills the SMF using a focused ion beam (FIB) to obtain a three-wave FPI. The three sets of interference fringes obtained allow the simultaneous measurement of ambient temperature and water salinity variations with normalized root mean square errors of 3.8% and 3%, respectively. In 2014, Que Ruyue [57] prepared a RI sensor based on a double-opening FPI cavity inside an optical fiber by using a femtosecond laser direct writing technique to etch a rectangular notch at the end of the fiber and combining it with an optical fiber fusion method. The temperature crosstalk was less than 0.0025 nm/°C, and the sensitivity of seawater salinity measurement was 0.171 nm/(mg/mL). Raquel Flores [58] used a focused ion beam to mill out microfluidic channels on the F-P cavity and used the Vernier effect to measure highly sensitive salinity and temperature dual parameters simultaneously, with a sensitivity of 6830.0 nm/RIU for salinity due to the amplification of the Vernier effect. In 2019, Jiuxing Jiang [59] obtained an open-cavity FPI by adjusting the lateral offset, allowing the liquid to be measured to enter and leave the cavity freely, and its fabrication process uses only cutting and fusion. In 2021, Hong-Kun Zheng [60] proposed a reflective fiber optic sensor consisting of two F-P cavities (Figure 6c). One of them is directly exposed to the environment to sense the ambient salinity, and the other one is used to compensate for the temperature coupling effect. In particular, they coated a thin gold film on the RI change interface so that the spectral quality was no longer limited by the RI difference between the two sides of the interface. Moreover, the frequency division multiplexing (FDM) technique and the cavity length demodulation technique were used to achieve full-scale salinity measurements. The sensitivity of the sensor was 50 nm/‰, and the sensor exhibited broad application prospects in salinity measurement.

The diaphragm in the FP cavity is a precision component. In 2015, Mingran Quan [61] proposed a microporous silver membrane to solve the problem that the liquid cannot flow into the cavity, which allows the FP cavity to be filled with the measured liquid, and its salinity measurement sensitivity reached 1025 nm/RIU. Xinpu Zhang [62] used a polyimide (PI) diaphragm. The PI diaphragm shrank with salinity, thus causing a redshift of the interference fringe. The maximum sensitivity of salinity measurement was 0.45 nm/(mol/L), indicating that this system has the advantage of not requiring calibration and ability to be used for real-time salinity sensing applications.

New solutions are also emerging in the structural design of OFSs. For example, Jian Zhao [63] combined anti-resonance (AR) with FPI. As shown in Figure 6d, its overall structure is a hybrid sensor with an SMF-HCF-NCF-SMF (SHNS) structure (Figure 6d). This solution has a salinity sensitivity of 0.235 nm/‰, and a modified FPI structure is available to improve the salinity measurement sensitivity. Unfortunately, the polymer coating will be affected by seawater and has limitations for long-term measurements. Similarly, Yong Zhao [64] used a capillary-type HCF as a microfluidic device to measure seawater salinity for the first time, and the corresponding minimum detectable resolution reached 0.0008 parts per thousand, indicating that it can be used for small-volume liquid salinity monitoring.

In general, FPIs have the advantages of a simple structure, high stability in most cases and the highest sensitivity index. Therefore, they show promise for application in the in situ measurement of seawater salinity.

**Figure 6 sensors-23-02187-f006:**
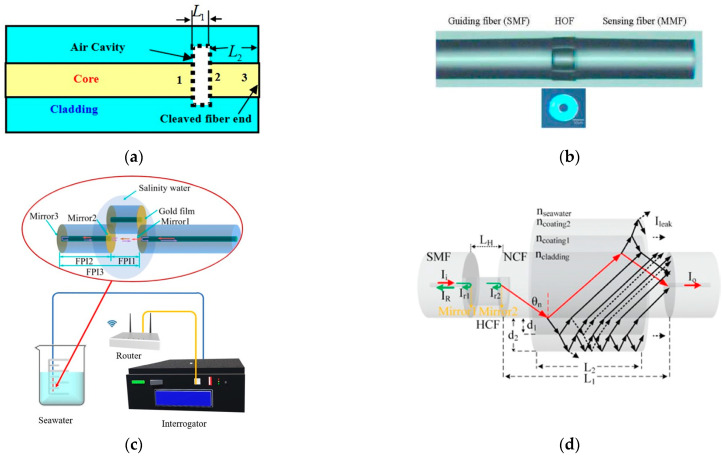
(**a**) Structure of the sensor probe [54]. (**b**) Microscope image of the fabricated FP sensor [55]. (**c**) Schematic of the proposed sensing structure [60]. (**d**) Diagram of the proposed SHNS sensing structure [63].

**Table 3 sensors-23-02187-t003:** Summary of FPIs.

Application	Year	Reference	Technique	Parameters	Others	Size (Diameter)
RI sensor	2008	[54]	FPI	RI resolution: ~4 × 10^−5^	wavelength: 1550 nm temperature: -	-
Salinity and Temperature sensor	2009	[55]	MMF-HOF and dual-cavity FPI	Sensitivity: 16 dB/RIU Range: 1.33 to 1.45	wavelength: 1450–1650 nm temperature: 26–500 °C	-
Salinity and Temperature sensor	2010	[56]	Three-wave FPI	Sensitivity: - range: 0 to 60 ppt	Wavelength: 1532–1544 nm temperature: 0–50 °C	-
Salinity sensor	2014	[57]	Double-openings FPI	sensitivity: 0.171 nm/(mg/mL) range: 1.333 to 1.377	wavelength: 1480–1560 nm temperature: 20–80 °C	-
Salinity and Temperature sensor	2019	[58]	Vernier-effect and FPI	sensitivity: 82.61 nm/M range: 0 to 0.297 M	wavelength: 1544–1552 nm temperature: 0–8 °C	-
Salinity and Temperature sensor	2021	[60]	Double-FPI	sensitivity: >50 nm/‰ range: 0‰ to 40‰	wavelength: 1550 nm temperature: 10–40 °C	-
RI sensor	2015	[61]	Microporous silver diaphragm FPI	sensitivity: 1025 nm/RIU range: -	wavelength: 1566–1578 nm temperature:	-
Salinity sensor	2015	[62]	FPI and PI	sensitivity: 1025 nm/RIU range: 0 to 5.47 mol/L	wavelength: 1540–1560 nm temperature: 30–80 °C	-
Salinity and Temperature sensor	2022	[63]	SMF-HCF-NCF-SMF (SHNS) FPI	sensitivity: 0.235 nm/‰ range: 0‰ to 40‰	wavelength: 1520–1580 nm temperature: 15–35 °C	-
Salinity and Temperature sensor	2022	[64]	FPI and AR	sensitivity: −1.152 nm/‰ range: 0‰ to 40‰	wavelength: 1520–620 nm temperature: 20–25 °C	-
RI sensor	2019	[59]	FPI and lateral offset splicing	sensitivity: 1013.8 nm/RIU range: 1.3464 to 1.3777	wavelength: 650–820 nm temperature: -	

#### 3.3.2. Sagnac Interferometer

A salinity OFS based on SI is mainly a single-parameter measurement, which is divided into two main categories. One relies mainly on the coating applied to the fiber. Similar to the FBG-based schemes, the variation in water salinity has an effect on the coating film, which generates axial strain or radial pressure on the fiber due to the dissolution effect. For example, related research was started earlier by Chuang Wu et al., who designed a high-sensitivity fiber optic salinity sensor using a PI-coated PM-PCF SI (Figure 7a) in 2011 [65]. The method exploits the fiber radial pressure effect of PM-PCF SI. Its salinity sensitivity is approximately 0.742 nm/(mol/L). This is the first time that the fiber coating-induced pressure effect has been used for salinity sensing. The obtained salinity sensitivity was 45 times higher than that of the PI-coated FBG. The disadvantage is that the response time was excessively long.

The other category is based on swift-wave detection under the condition of the RI-salinity conversion relationship. This is also a recent mainstream scheme. In 2014, Chuang Wu [66] et al. combined SI-based swift-wave detection with microfiber technology to fabricate microfibers using the flame heating stretching technique. The microfiber has a rectangular cross-section with a width of 4.0 μm × 2.5 μm and a total length of 36 mm. It has a very strong swift wave field and is therefore very sensitive to changes in the surrounding RI. For water salinity in the range of 0‰ to 40‰, the sensitivity is 1.95 nm/‰, and the detection limit is 0.01‰. In 2020, Md. Aslam Mollah [67] made some improvements in combination with PCF. The cross section of the proposed salinity sensor is shown in Figure 7b, which indicates the field distribution of x-polarization and y-polarization, respectively. Three air holes were removed from the central part to form the core of the PCF. The sensitivity obtained reached 7.5 nm/% in the salinity range from 0% to 100%. The maximum resolution was 2.66 × 10^−6^ RIU. This scheme had a relatively simple structure, high sensitivity and a high potential for seawater salinity measurements. In 2021, Yang et al. [68] proposed a tapered PMF-SI (Figure 7c) for simultaneous measurement of the salinity and temperature of seawater. They investigated the effect of the distance between PMF cones and the fiber cone diameter on the sensor performance. The experimental results showed that a cone waist diameter of 20 μm and a distance of 3 cm were optimal for a salinity sensitivity of 0.367 nm/%.

SI systems mostly involve single-parameter measurements, which is its main disadvantage as a salinity sensor. In the previous section, we mentioned that both temperature and pressure affect the salinity measurement, so if SI is needed as an in situ measurement solution, it is preferred to integrate other sensors for the measurement.

**Figure 7 sensors-23-02187-f007:**
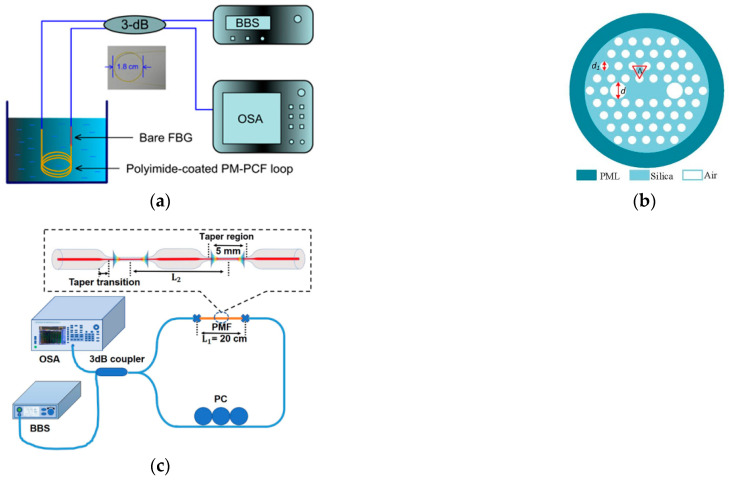
(**a**) Schematic diagram of the PM-PCF SI salinity measurement with the output spectrum of the sensor probe [65]. (**b**) The proposed seawater−filled PCF [67]. (**c**) Schematic diagram of the performance test system for the proposed SI based on concatenated panda PMF tapers [68]. Reprinted with permission from Ref. [68] © The Optical Society.

**Table 4 sensors-23-02187-t004:** Summary of SIs.

Application	Year	Reference	Technique	Parameters	Others	Size (Diameter)
Salinity sensor	2011	[65]	PI-coated PMPCF SI	sensitivity: 0.742 nm/(mol/L) range: 0 to 5 mol/L	wavelength: 1500–1600 nm temperature: -	1.8 cm
Salinity sensor	2014	[66]	Microfiber SI	sensitivity: 1.95 nm/‰ range: 0‰ to 40‰	wavelength: 1200–1800 nm temperature: -	-
Salinity sensor	2020	[67]	PCF SI	sensitivity: 0.75 nm/‰ range: 0% to 100%	wavelength: 1000–2500 nm temperature: -	-
Salinity and Temperature sensor	2021	[68]	Tapered PMF-SI	sensitivity: 0.367 nm/% range: -	Wavelength: 1520–1540 nm temperature: 25–55 °C	20 cm (PMF)

#### 3.3.3. M-Z Interferometer

One more type of fiber optic interferometer is MZI. In 2012, C. Gouveia et al. [69] proposed a differential MZI system with a sensitivity of up to 57,404°/RIU and a minimum detectable RI change of ±2 × 10^−6^. Unfortunately, the scheme has not been applied to salinity measurements but has provided helpful information that can be used for the application of the MZI to seawater salinity measurements. In 2016, Kumari et al. [70] applied the MZI to produce seawater salinity measurements and used a femtosecond laser as the light source with a salinity range of 31 to 37 ppt at a constant temperature of 27 °C with a sensitivity of 0.01 μm/ppt.

According to the TEOS-10 equation, seawater salinity is closely related to temperature, and in the same year, Yipeng Liao [71] et al. proposed a new method to measure salinity and temperature simultaneously using a microfiber MZI with a knot resonator (MZIKR) with a salinity sensitivity of 208.63 pm/‰. Additionally, André D. Gomes and Orlando Frazão [72] used the microfiber knot resonator the following year together with an abrupt taper-based MZI (Figure 8a) to improve the sensitivity of salinity measurements with simultaneous temperature compensation based on the advantages of the dual parameter, with a sensitivity of 1354 nm/RIU in the RI range 1.32823 to 1.33001. However, this solution does not directly measure the salinity of the liquid and must be tested or converted to salinity for field applications.

Microfibers have a large number of applications in addition to MZIKR scheme combinations. For example, in 2018, Nanjie Xie et al. [73] used an in-line microfiber-assisted MZI (MAMZI) scheme, with its interference arms constituted by a microfiber of a few hundred micrometers in length and a U-shaped microcavity, as shown in Figure 8b, which was constructed by splicing a section of a single-mode fiber (SMF) of a few hundred micrometers in length between two SMFs that gradually tapered from the microfiber. Due to this structure, the salinity sensitivity of the method was as high as 2.419 nm/‰. In 2019, Tianqi Liu [74] et al. proposed a splicing point tapered fiber MZI, a scheme combined with nonadiabatic tapering and mode field mismatch between two different fibers. The unique design of the MZI after theoretical analysis allowed us to obtain the transmission spectrum of dual-parameter sensing with two sets of clear interferences, as shown in Figure 8c. The highest sensitivity of salinity was up to 0.29047 nm/‰. The method is highly practical as a two-parameter sensing device that can be easily fabricated and packaged.

In terms of sensor structure, Wang, Shanshan [75] et al. proposed an all-fiber hybrid structure MZI based on silica fiber and fluorinated polyimide microfiber (FPMF) for the temperature or salinity sensing of seawater. A salinity sensitivity of 0.064 nm/‰ was obtained. The exposed-core microstructured optical fiber (ECF) structure can be used to enhance the sensitivity. In 2019, Lina Wang [76] proposed a high-sensitivity salinity sensor based on a free-space propagation core mode Mach–Zehnder interferometer, benefiting from the exposed-core microstructured fiber structure feature (Figure 8d). As shown in Figure 8e, the overall structure is an SMF-ECF-SMF structure with a salinity sensitivity of approximately −2.29 nm/‰.

More recent schemes for splicing different structural fibers similar to that mentioned above have also emerged. In 2021, Ziting Lin [77] et al. proposed an open-cavity MZI structural sensor with an open-cavity structure consisting of a small section of etched double-sided hole fiber that is spliced between a pair of multimode fibers and cascaded in a pair of single-mode fibers, i.e., the proposed SMF-MMF-etched DSHF-MMF-SMF structure. The salinity sensitivity of the probe reached 2 nm/‰, and the RI sensitivity was greater than 10,000 nm/RIU; the probe also exhibited low loss and a salinity detection limit of 0.23‰. This system has low fabrication costs and simple steps compared with the femtosecond inscribing method. In the same year, Jinmeng Yan [78] made some variations on the above scheme, using open-cavity MZI structures in combination with no-core fiber (NCF) and achieved a sensitivity of up to 3.444 nm/% at 0.5–5% salinity. Tanushree Selokar [79] developed balloon-shaped SMF structures and core diameter mismatch (CDM) structures for MZI temperature and salt sensors with a sensitivity of 168.35 pm/% for salinity measurements in the range of 5–35%. Zi-ting Lin et al. [80] proposed a highly sensitive MZI salinity sensor with the main structural composition of SMF-(C-type microstructured optical fiber) CMOF-MMF-SMF (Figure 8f), and the salinity sensitivity of the sensing probe was up to 3.25 nm/‰ in the salinity range of 0–40‰. In addition, the sensing probe exhibited a relatively good detection limit of 0.1‰, surpassing the 1‰ index in the primary scheme of conductivity salinity measurement compared to the conductivity scheme. In the same year, the same group [81] continued to focus on the MZI of C-type microstructured optical fibers combined with a single mode fiber (SMF)-no-core fiber-double-C fiber (DCF)-NCF-SMF structure by etching double-sided hole fibers with HF acid to prepare DCFs. The large size of the exposed microfluidic channels of DCFs overcame the challenging liquid filling and replacement problem experienced by previous microstructured fibers, which have a salinity range of −2.26 nm/‰ salinity sensitivity.

In addition to the structure, in 2019, Rende Ma [82] first split the wave front of a Gaussian beam with a polyethylene terephthalate (PET) film and then focused it into the SMF by another collimator to form a stable fiber MZI. Its salinity sensitivity can reach −0.81 nm/‰. Based on the phase compensation method, the sensitivity of this scheme can be further enhanced. In 2022, Xiaoping Li [83] found that dispersion will play a key role in the sensitization technique of RI. The experimental results show that the sensitivity reaches −1.467 × 10^5^ nm/RIU when the salinity was 1‰. In addition to this method, greater sensitivity can be obtained by matching the RI and thickness of the wavefront separator with those of the phase compensator.

In general, MZIs have been used for sensing measurements of a large number of parameters, especially temperature and salinity sensors, and can be applied to multiparameter sensing. MZIs also have good prospects for in situ measurement applications.

**Figure 8 sensors-23-02187-f008:**
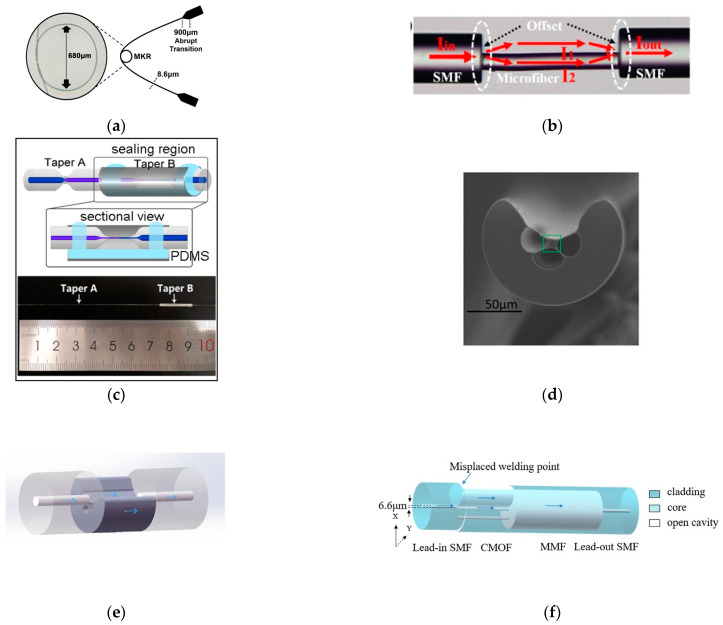
(**a**) Schematic of the proposed sensor and picture of the microfiber knot resonator [72]. (**b**) Microscopic image of the fabricated MAMZI [73]. (**c**) Schematic of the capsulation method and a photo of the MZI after sealing. Transmission spectra of MZI [74]. (**d**) Cross-section view of the ECF [76]. (**e**) Schematic of the sensor based on the SMF-ECF-SMF structure [77]. (**f**) Configuration of the SMDMS sensing structure [80]. Reprinted with permission from Ref. [77] © The Optical Society.

**Table 5 sensors-23-02187-t005:** Summary of MZIs.

Application	Year	Reference	Technique	Parameters	Others	Size (Diameter)
RI sensor	2012	[69]	Tapered MZI	sensitivity: 57,404°/RIU resolution: 2 × 10^−6^ range: 1.3355 to 1.3485	wavelength: - temperature: -	-
Salinity sensor	2016	[70]	Femtosecond laser and MZ	sensitivity: 0.01 mu/ppt range: 31 to 37ppt	wavelength: 600–1700 nm temperature: 27 °C	-
Salinity and Temperature sensor	2016	[71]	Microfiber MZIKR	sensitivity: 0.75 nm/‰ range: 0% to 100%	wavelength: 1508–1520 nm temperature: 13.7–25.0 °C	960 μm
RI and Temperature sensor	2017	[72]	Microfiber MZIKR	sensitivity: 1354 ± 14 nm/RIU range: 1.32823 to 1.33001	wavelength: 1525–1545 nm temperature: 0–44 °C	680 μm
Salinity sensor	2018	[73]	MAMZI	sensitivity: 2.419 nm/‰ range: 0 to 3 wt%	wavelength: 1300–1600 nm temperature: -	50 μm
Salinity and Temperature sensor	2019	[74]	Splicing point and Tapered MZI	sensitivity: 290.47 pm/‰ range: 0 to 3 wt%	wavelength: 1550 nm temperature: 22.3 °C	10–30 μm
Salinity and Temperature sensor	2018	[75]	SiO_2_ fiber and FPMF MZI	sensitivity: 64 pm/‰ range: 35‰ to 54‰	Wavelength: around 1535 nm temperature: 20.3–29.2 °C	5.52 μm
Salinity sensor	2019	[76]	SMF-ECF-SMF MZI	sensitivity: −2.29 nm/‰ range: 0‰ to 40‰	wavelength: 1400–1700 nm temperature: 25–85 °C	-
Salinity sensor	2021	[77]	Open-cavity MZI	sensitivity: 2 nm/‰ range: 0‰ to 40‰	wavelength: 1300–1700 nm temperature: 20–45 °C	90 μm
Salinity sensor	2021	[78]	NCF open-cavity MZI	sensitivity: 3.444 nm/% range: 0.5–5%	wavelength: 1540–1600 nm temperature: 20–40 °C	62.5 μm
Salinity and Temperature sensor	2021	[79]	Balloon shaped SMF and CDM MZI	sensitivity: 168.35 pm/% range: 5% to 35%	Wavelength: 1500–1630 nm temperature: 23 °C	-
Salinity sensor	2022	[80]	SCMS MZI	sensitivity: −3.25 nm/‰ range: 0‰ to 40‰	Wavelength: 1500–1650 nm temperature: -	63.5 μm (corroded cavity)
Salinity sensor	2022	[81]	SMF-NCF-DCF-SMF MZI	sensitivity: −2.26 nm/‰ range: 10‰ to 50‰	wavelength: 1450–1600 nm temperature: 16 °C	96.5 μm
Salinity sensor	2019	[82]	MZI	sensitivity: −0.81 nm/‰ range: 0.5 to 5%	wavelength: 1100–1600 nm temperature: -	-
RI sensor	2022	[83]	WFSF-MZI	sensitivity: −1.467 × 10^5^ nm/RIU range: -	wavelength: 1260–1650 nm temperature: 20–30 °C	-

### 3.4. Hybrid OFS

Most of the various schemes mentioned above are based on individual techniques, with some researchers focusing on improving the sensitivity of the sensors and others exploring high-resolution demodulation methods. In recent years, a number of hybrid OFS schemes have been proposed. They usually have the characteristics of a single scheme but also complement each other’s strengths.

#### 3.4.1. Hybrid FPI

FP cavities are relatively simple in structure and easy to combine with other schemes.

The scheme of FPIs mixed with MZIs is more popular. In 2020, Hongkun Zheng et al. [84] divided light into multiple parts at the offset interface, where the transmitted light formed the MZI spectrum, and the reflected light formed the FPI spectrum (Figure 9a). Thus, the dual parameters of salinity and temperature could be measured separately, and the coupling effect was effectively eliminated. The salinity sensitivity of this scheme was 2.4473 nm/‰ in the range of 20–40‰. This scheme makes full use of the respective advantages of the two interferometers.

Professor Yong Zhao’s group at Northeastern University has proposed many innovative solutions in the form of hybrid structures. In 2021, Hong-Kun Zheng [85] et al. used a reflective fiber probe processed with a special sputtering technique. As shown in Figure 9b, they integrated two interferometers in the same channel and separated the spectrum by FDM. The experimental results show that the salinity sensitivity was up to 2.7 nm/‰, and this reflective probe improved the usefulness of the sensor. The following year, they also attempted to develop a UV adhesive processing scheme for the above structure [86] (Figure 9c), but the UV adhesive could not maintain long-term stable operation in seawater. Notably, they recently proposed a noteworthy structural approach [87]. As shown in Figure 9d, using a section of the HCF spliced between SMFs to form the FPI and a U-shaped slot inscribed in the HCF with an Fs laser, the transmitted light in the core was coupled to the laser inner cut waveguide to form the MZI. This design could determine seawater salinity by relating it to the resonance wavelength, and the overall structure is compact, sensitive and reproducible. In addition, Sema Guvenc Kilic [88] proposed a FPI formed by two chirped fiber Bragg gratings on a seven-core multicore fiber and used to measure RI, with a coaxial FBG used for temperature compensation to improve real-time performance. Overall, hybrid FPI schemes are widely proposed thanks to the simple and flexible structure of FPI.

Hybrid FPIs can use each sensing mechanism for simultaneous multiparameter sensing measurements on the one hand and can also take advantage of the Vernier effect for sensitivity enhancement on the other hand. Unfortunately, most hybrid FPIs are currently in the laboratory validation stage and need to be further validated before use for in situ measurements in a marine environment.

**Table 6 sensors-23-02187-t006:** Summary of hybrid FPIs.

Application	Year	Reference	Technique	Parameters	Others	Size (Diameter)
Salinity and Temperature sensor	2020	[84]	FPI-MZI	sensitivity: 2.4473 nm/‰ range: 20‰ to 40‰	wavelength: 1460–1620 nm temperature: 28–48 °C	-
Salinity and Temperature sensor	2021	[85]	FPI-MZI	sensitivity: 2.7 nm/‰ range: 0‰ to 40‰	wavelength: 1460–1620 nm temperature: 10–40 °C	-
Salinity and Temperature sensor	2022	[86]	FPI-MZI	sensitivity: 2.323 nm/‰ range: 0‰ to 40‰	wavelength: 1400–1600 nm temperature: 5–45 °C	-
Salinity and Temperature sensor	2022	[87]	FPI-MZI	sensitivity: 0.244 nm/‰ range: 0‰ to 40‰	wavelength: 1540–1600 nm temperature: 20–32 °C	-
RI and Temperature sensor	2019	[88]	FPI-FBG	sensitivity: 1.43 nm/RIU range: 1.316	wavelength: 1550 nm temperature: -	125 μm

#### 3.4.2. Hybrid FBG

FBG solutions also have more hybrid structure application cases, which are partially exemplified here.

FBG cascaded with the MZI: In 2018, Niu et al. [89] combined an FBG with an S fiber taper (SFT)-MZI (Figure 10a) to convert wavelength demodulation to intensity demodulation with the aid of FBG reflection. The RI sensitivity was up to 366.69 dB/RIU, and the temperature complement requirement was low.

FBG cascaded with LPFG: In 2020, Bo Pang [90] et al. designed a temperature and RI dual-parameter sensor by exploiting the different characteristics of LPFG and FBG for temperature and SRI response (Figure 10b). The sensitivity of the sensor to temperature and SRI reached 10 pm/°C and 2326.7 nm/RIU, respectively.

FBG cascaded with SI: In 2021, SUN [91] proposed an FBG hybrid microfiber Sagnac ring RI sensor for measuring the RI of liquids. The SMF used was almost etched into the fiber core, cascaded with the FBG and connected to a Sagnac fiber ring structure. The results show that the sensor had a RI sensitivity of 1787.4 nm/RIU.

FBG cascaded with the BFI: First developed in 2022, this scheme [92] cascades the BFI with a PMFBG (Figure 10c), where two different kinds of PM optical fibers spliced with a 45° rotation between their polarization axes are composed of BFI. The dip wavelength shifts are related to the surrounding strain and temperature, and the system detects the RI using a Michelson interferometer (MI) implemented by the waist amplification technique. This sensor benefits from the cascade structure and can simultaneously measure three parameters: temperature, strain and salinity.

Hybrid FBGs have advantages similar to those of hybrid FPIs, but in general, a hybrid OFS is usually more complex in terms of processing and structure.

**Figure 10 sensors-23-02187-f010:**
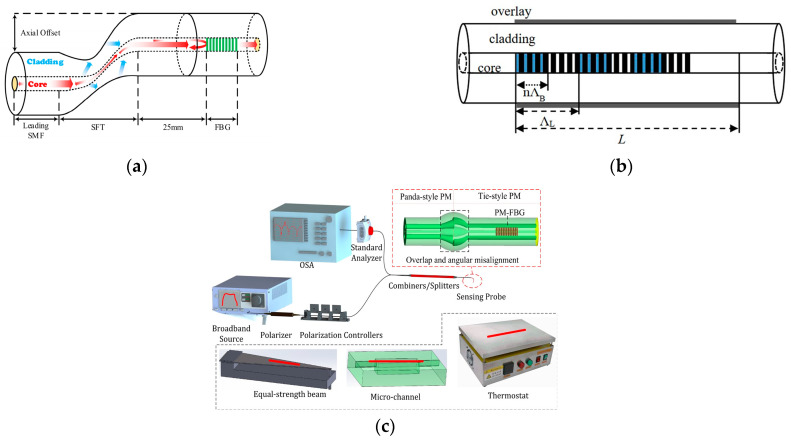
(**a**) Schematic diagram of the proposed RI sensor [89]. (**b**) Superimposed coated LPFG and FBG structures [90]. (**c**) Schematic configuration and experimental setup of the sensing probe [92].

**Table 7 sensors-23-02187-t007:** Summary of hybrid FBGs.

Application	Year	Reference	Technique	Parameters	Others	Size (Diameter)
RI sensor	2018	[89]	FBG-SFT-MZI	sensitivity: 366.69 dB/RIU range: 1.3330 to 1.3988	wavelength: 1582–1587 nm temperature: 30–70 °C	-
RI sensor	2020	[90]	SCLBG	sensitivity: 2326.7 nm/RIU range: 1.33 to 1.34	wavelength: 1500–1800 nm temperature: 20–70 °C	-
RI sensor	2021	[91]	FBG and microfiber Sagnac ring	sensitivity: 1787.4 nm/RIU range: 1.333 to 1.369	wavelength: 1540–1610 nm temperature: -	-
Salinity, Temperature, and strain sensor	2022	[92]	PM-FBG and BFI	sensitivity: 70.82 nm/RIU range: 1.3373 to 1.3415	wavelength: 1260–1625 nm temperature: 34–46 °C	-

### 3.5. Microfibers

Microfibers are a current research hotspot. Some of the methods summarized above already include microfiber technology, and some additions are made here. Recently, microfiber-based multimodal interferometers operating near the dispersion turning point (DTP) have been investigated and serve as promising candidates for liquid sensing due to their ultrahigh sensitivity [93,94,95,96]. Some researchers have decreased the microfiber diameter to enhance the sensitivity of RI measurements. Shengyao Xu [97], on the other hand, optimized this structure. As shown in Figure 11, they proposed an ultrasensitive enhanced fabrication-tolerance refractometer using the polarization interference of a tapered PANDA-air-hole fiber (PAHF). The polarization interference of the tapered PAHF, the tapered profile of this PAHF-based microfiber, is due to the tunable birefringent dispersion. The sensor structure significantly extends the salinity measurement range from 1. 3324 to 1.3328 in the ultrasensitive band at 600 nm with a salinity sensitivity of 47223 nm/RIU.

Additionally, in the study of low-volume liquids, in 2020, Tinko Eftimov et al. [98] proposed the application of LPGs and μIMZI for DTP-LPG using etching and nanocoating to improve the sensitivity. RI units up to 20,000 nm/RIU for DTP LPGs and up to 27,000 nm/RIU for μIMZI were achieved. Although DTP LPG offers greater interaction lengths and sensitivity to changes occurring on their surface, μIMZI is more suitable for the study of small volumes of liquids.

In particular, an optical microfiber coupler (OMC) [99] not only has swift field transmission characteristics but also has high sensing sensitivity. It has the advantages of easy fabrication, compact structure, high sensitivity, low cost and compatibility with fiber optic systems and has a great potential for deep-sea applications.

In general, microfiber-based OFSs have the advantage of high sensitivity, but due to their smaller diameter and greater fragility, they also require more packaging in applications.

**Table 8 sensors-23-02187-t008:** Summary of microfibers.

Application	Year	Reference	Technique	Parameters	Others	Size (Diameter)
RI sensor	2021	[97]	PAHF-based microfiber	sensitivity: 47223 nm/RIU range: 1.3324 to 1.3328	wavelength:1550 nm temperature: 25 °C	3.6 μm
RI sensor	2020	[98]	DTP-LPG and μIMZI	sensitivity: 27000 nm/RIU range: -	Wavelength: 1583 nm temperature: -	50–60 μm
Salinity and Temperature sensor	2018	[75]	SiO_2_ fiber and FPMF MZI	sensitivity: 0.064 nm/‰ range: 35‰ to 54‰	Wavelength: around 1535 nm temperature: 20.3–29.2 °C	5.52 μm
Salinity, Temperature, and Depth sensor	2018	[99]	OMC	sensitivity: 1.596 nm/‰ range: 0‰ to 45‰	wavelength: 1525–1560 nm temperature: 25.1 °C	-

## 4. Comparison and Future of Salinity OFSs

### 4.1. Comparison of Different OFSs

A comparison of the performance of various sensors indicates that the basic sensitivity of the fiber optic SPR type salinity sensor is higher, but due to the poor corrosion resistance of Au films, long-term in situ seawater measurements cannot currently be achieved. The basic sensitivity of the fiber grating type salinity sensor is poor, but the stability is strong, which can satisfy some long-term requirements and does not require high-resolution marine environment detection. In addition, optical fiber grating-type salinity sensors with integrated multiplexing have certain advantages that make them conducive to large-scale networking. More types of interferometric OFS salinity sensors and hybrid OFSs are available due to different combinations, and they exhibit different characteristics and the overall advantages of flexibility and high sensitivity. Microfibers, as new micro-optical devices, have a high sensitivity based on their unique swift wave transmission characteristics and have excellent future application prospects in marine environment monitoring. Please refer to Table 9 for specific advantages and disadvantages.

Finally, to make a comprehensive comparison of different measurement methods, Table 10 was created to compare different sensors from the aspects of sensitivity, Measurement range, fabrication difficulty, and stability. Fabrication difficulty represents the manufacturing difficulty of the processing of the sensor and the manufacturing process for sensors can often be very complex, so only the most basic processing methods are listed here. Stability represents the ability of the sensor to operate in seawater over a long period of time, and as most technical solutions have not been tested in the sea, they can only be judged by the construction and materials they use.

**Table 9 sensors-23-02187-t009:** Summary of the advantages and disadvantages of the different methods.

Structure	Advantage	Disadvantage
Optical fiber SPR	High sensitivity	Poor corrosion resistance of the coated metal film
Optical fiber grating	Stable for long periods in seawater	Low base sensitivity
Interferometric OFS	High structural flexibility and sensitivity	No proven marine test cases, further research needed in encapsulation
Hybrid OFS	Flexible construction, salinity spikes in temperature compensation can be solved by a co-axial cascade solution	Most of them are currently in the laboratory testing phase and their long-term stability in the ocean needs to be further proven.
Microfiber	Low loss and excellent sensitivity	The structure is relatively fragile compared to the sea trial and need armored during the sea trial

**Table 10 sensors-23-02187-t010:** Comparison of different OFSs.

Type	Structure	Sensitivity (nm/‰)	Measurement Range (‰)	Fabrication Difficulty	Stability
SPR		0.00558 [25]–0.445 [29]	0–40	Difficult (silver mirror reaction method)	Low
FBG		0.0104 [43]–0.0165 nm/M * [44]	0–40	Medium (Chemical etching)	High
Interferometric OFS	FPI SI MZI	0.171 nm/(mg·mL^−1^) * [57]–>50 [60] 0.367 [68]–0.75 [67] 0.064 [75]–3.444 [78]	0–50	Medium (etching, fusion splicing, and femtosecond laser)	Depends on the specific structure
Hybrid OFS	FPI MZI FBG MZI FBG SI FBG LPFG	0.244 [87]–2.7 [85] 366.69 dB/RIU * [89] 1787.4 nm/RIU * [91] 2326.7 nm/RIU * [90]	0–40 1.3330–1.3988(RIU) * 1.333–1.369(RIU) * 1.33–1.34(RIU) *	Medium (etching, fusion splicing, and femtosecond laser)	Depends on the specific structure
Microfiber	MZI OMC	0.064 [75] 1.596 [99]	35–45 0–45	Difficult (direct drawing process and flame-brushing method)	Depends on the specific structure

* Units are inconsistent and have been marked.

### 4.2. Future Research and Development Direction

Most of the above-summarized technologies are currently only demonstrated in the laboratory, but an OFS salinity sensor must be used in marine applications to achieve “what you measure is what you get”. First, compared to the CTD instruments of Sea Bird in the U.S., which have been applied to marine environmental monitoring, some of the sensors need to be replenished with liquid during the calibration process, which requires structure optimization. Second, the measurement standard needs to be unified. Most of the current measurement samples are NaCl solutions prepared in the laboratory, and this simple salt solution cannot effectively replicate the complex composition of seawater. Therefore, it is recommended to use standard seawater for calibration measurements in subsequent studies. Unlike the conductivity method used to obtain practical salinity, OFS salinity sensors based on the RI measurement principle can directly obtain absolute salinity. Third, environmental adaptability should be enhanced. In marine environment monitoring, it is necessary to enhance the response to the problem of marine pollutants, algae, and the adhesion of microscopic organisms, and the sensor needs to be both pressure- and corrosion resistant, which imposes strict requirements for sensor packaging (encapsulation design). In addition, integration and arrays are the current development trend of marine environmental monitoring. For OFS salinity sensors, further optimization of multiparameter cross-sensitivity and signal multiplexing demodulation methods is needed.

## 5. Conclusions

This paper presents the progress of currently available research on fiber optic salinity sensors for seawater based on the RI measurement principle. Using a fiber optic sensor to measure the RI of seawater to determine salinity is a more direct method to obtain the absolute salinity (TEOS-10 recommendation). This article compares and analyzes the advantages and disadvantages of different types of research methods, including optical fiber SPR, optical fiber grating, interferometric OFS, hybrid OFS, and microfiber. Finally, this paper discusses the current difficulties that need to be overcome and the outlook for OFS salinity sensors as they develop further toward marine environmental monitoring applications.

## Figures and Tables

**Figure 1 sensors-23-02187-f001:**
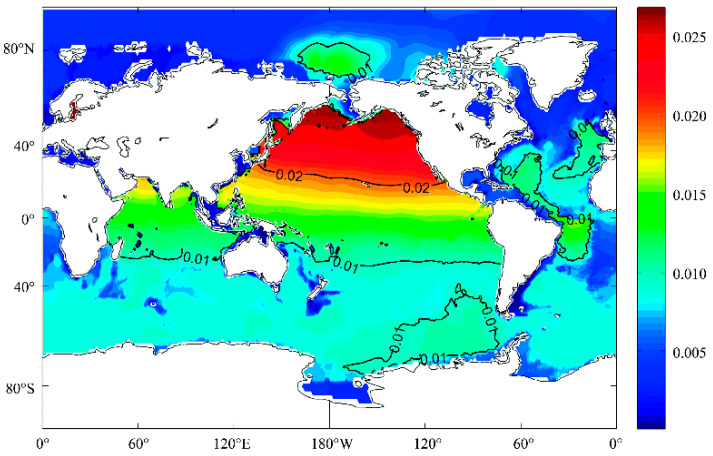
Maximum *δS_A_* in the world oceans (Data source: World Ocean Atlas 2009).

**Figure 2 sensors-23-02187-f002:**
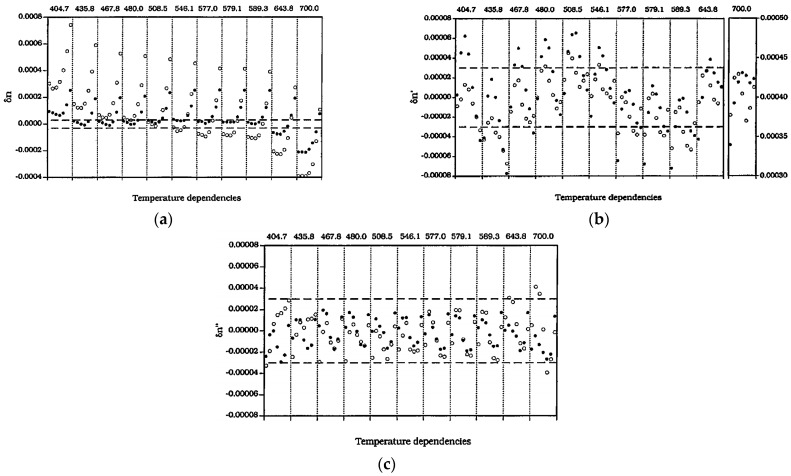
3 Empirical formula error distribution [22]. (**a**) The first equation. (**b**) The second equation. (**c**) The third equation.

**Figure 3 sensors-23-02187-f003:**
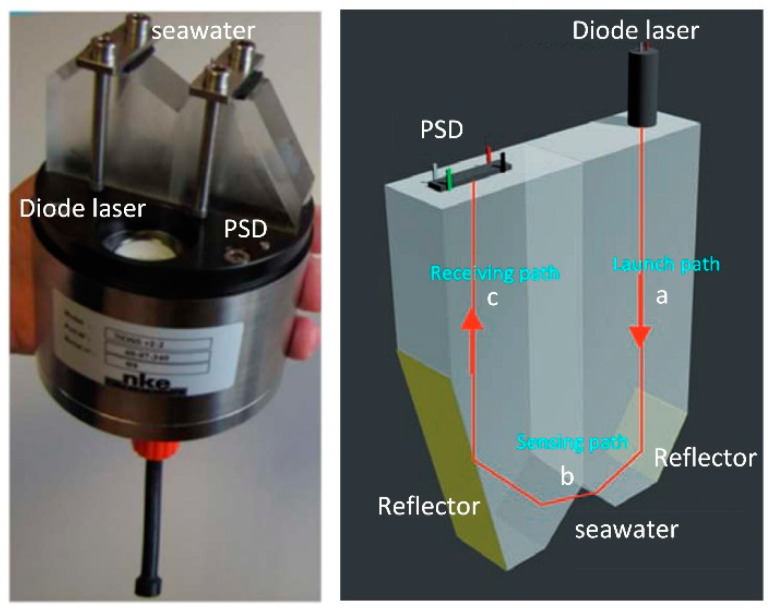
Principle and physical diagram of the RI salinity sensor from NKE, France [23].

**Figure 9 sensors-23-02187-f009:**
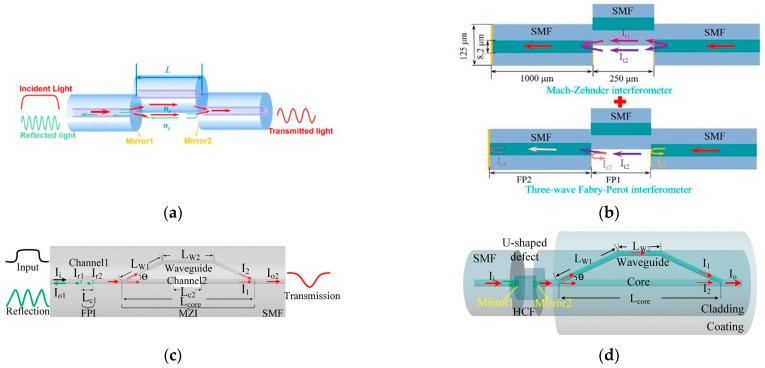
(**a**) Diagram of the proposed sensing structure [84]. (**b**) Light transmission path in the proposed sensing structure [85]. (**c**) Schematic of the proposed sensor [86]. (**d**) Diagram of the cascaded sensing structure [87]. Reprinted with permission from Ref. [84] © The Optical Society.

**Figure 11 sensors-23-02187-f011:**
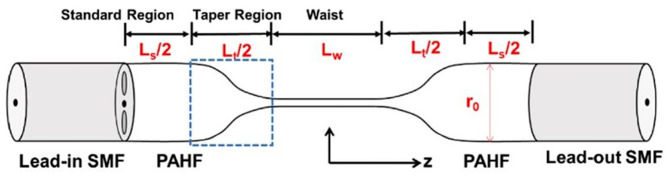
Schematic diagram showing the taper profile of the PAHF-based microfiber [97].

**Table 1 sensors-23-02187-t001:** Summary of optical fiber SPR sensors.

Application	Year	Reference	Technique	Parameters	Others	Size (Diameter)
Temperature and salinity sensor	2019	[25]	SPR and PDMS	sensitivity: 0.0558 nm/% range: 0 to 40‰	Wavelength: 400–1000 nm temperature: 20–40 °C	-
Temperature and salinity sensor	2020	[26]	SPR and PDMS, HCF	sensitivity: 0.3769 nm/‰ range: 0 to 40‰	Wavelength: 400–800 nm temperature: 20 °C	670 μm
Temperature and salinity sensor	2021	[27]	SPR and PDMS, PCF	sensitivity: 0.560 nm/‰ range: 0 to 40 ‰ resolution: 0.3257‰	Wavelength: 600–1000 nm temperature: 25 °C	-
Temperature and salinity sensor	2019	[28]	SPR and PDMS, PCF	sensitivity: 0.560 nm/‰ range: 0 to 40 g/kg	Wavelength: 400–800 nm temperature: 20 °C	125–188 μm
Temperature and salinity sensor	2021	[29]	SPR and EC-MOF	sensitivity: 0.445 nm/‰ range: 0 to 25%	wavelength: 500–1100 nm temperature: 10–50 °C	125 μm
RI sensors	2020	[34]	SPR and Au-TiO_2_	sensitivity: 9790 nm/RIU range: 1.36410 to 1.37175	Wavelength: 400–800 nm temperature: -	600 μm
RI, temperature sensor	2020	[35]	SPR and Au film, PDMS	sensitivity: 0.31 nm/‰ range: 1.36410 to 1.37175	wavelength: 400–800 nm temperature:	-
RI sensor	2020	[30]	SPR and PCF	sensitivity: 13,000 nm/RIU range: 1.33 to 1.37	Wavelength: 500–1000 nm temperature: -	-
RI sensor	2020	[32]	SPR and dual-core PCF	sensitivity: 5500 nm/RIU range: 1.33 to 1.37	Wavelength: 1000–1400 nm temperature: -	-
RI sensor	2021	[31]	SPR and PCF	sensitivity: 32,000 nm/RIU range: 1.33 to 1.37	Wavelength: 650–820 nm temperature: -	-
RI sensor	2022	[33]	SPR and dual-core D-Formed PCF	sensitivity: 18,800 nm/RIU range: 1.40 to 1.45	Wavelength: 500–1100 nm temperature: -	-

## Data Availability

Not applicable.

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
