# Peer review of "(untitled)"

_sensors, 2023, doi:10.3390/s23042187_

Round 1

Reviewer 1 Report

The manuscript titled Review of seawater fiber optic salinity sensors based on the refractive index detection principle presents the progress of currently available research on fiber optic salinity sensors for seawater based on the refractive index measurement principle. The author compares and analyzes the advantages and disadvantages of different types of research methods, including optical fiber SPR, optical fiber grating, interferometric OFS, hybrid OFS, and microfiber, and then discusses the current difficulties that need to be overcome and the outlook for OFS salinity sensors as they develop further toward marine environmental monitoring applications. This article deserves to be published after revisions. My suggestions are as follows:

1. The manuscript introduces many different methods, like optical fiber SPR, optical fiber grating, interferometric OFS, hybrid OFS, and microfiber, but lacks a summary description. So I suggest the author add a table to summarize the structure, principle, advantages and disadvantages of different methods.

2. The manuscript lacks explanation of the overall content of the tables listed.

3. What is the meaning of δSA in Figure 1., and I think the unit g/kg is meaningless.

4. There are two Table 5 in this manuscript, one is Summary of MZIs, and the other is Summary of Hybrid FPIs. The author should revise.

5. Figure 2 is not clear enough.

6. Some details need attention, like “ In 2020, Wen Li [29] performed a theoretical analysis of an oxide (MgO, TeO2, and TiO2) fiber optic SPR sensor on Au film (Figure 4d). TeO2 and TiO2 should be written TeO2 and TiO2.

7. Wrong chapter number. 3.2 Optical grating, but following is “3.1.1. FBG”, “3.1.2. LPG”.

8. For a review, the number of references is still insufficient. Authors should refer to more recently published articles and compare them with previous studies. Based on these, the research status, technology improvement and development trend in this field are described.

Reviewer 2 Report

Dear Secretariat of Sensors Journal

Good morning and I hope you are doing well.

I would like to thank the Editor for entrusting me to review the paper entitled: Review of seawater fiber optic salinity sensors based on the re- 2 fractive index detection principle. After perusing the manuscript, I would like to provide some comments to the manuscript. Please refer to the detailed comments below. In addition, for general comments, I would like to propose the following inputs to be taken into account. Please find the general comments below:

1.       Please request English master to improve the English quality.

2.       Please also add additional relevant cited literatures to support your study, and please make the more logic

3.       Please insert the novelties of the study, you can add the novelties in the introduction part and make them in line with your study background

4.       Please also update the cited literatures in the reference part of your study. Please consider to take about the last-10 year citations.

5.       Almost 80% researchers cited to provide data based on Table 1-7 are from China, it will be grateful if you also diverse the cited researchers.

Thank you so much and I am looking forward to hearing your revision soon.

1.       Abstract: The authors are suggested to provide a brief method about the literature review used. Please also specify what kinds of reviews did you carry out, such as traditional review, systematic review, meta analysis or what?

2.       Abstract: Sentence 1 and Sentence 2 are not in line,

3.       Please also provide the objectives of your study

4.       What is TEOS abbreviation?

5.       What is OFSs abbreviation?

6.       Introduction: Line 41-42, please provide min values of high temperature and high salt

7.       What are the drawbacks of traditional methods to mesure seawater saliniy: weighing, chlorinity titration, and conductivity? You only stated the info of conductivity method

8.       In the salinity range of 0-50‰,? Percentage?

9.       Figure 1. How do you get the Fig?

10.   Introduction: Please provide some histories of the salinity measurement using refractive index measurement-based sensors. Please provide brief information about the histories, and compare with the additional methods

11.   Where is Section 1?

12.   Please provide abbreviations part. The readers will be difficult to understand the abbreviations if no long words explanation

13.   Please provide a brief explanation about the method did you use

14.   Principle: Please make the short of the words you abbreviated more clear and consistent. Such as capital letter or not, ect.

15.   Please provide the unit of the parameters did you use for the explanations

16.   Figure 3, please provide the parts of the tools. You can use letter a, b, c, etc. And its functions. Figure 3 should be better to provide the mechanism

17.   Please briefly summarize based on structure, principle, advantages and disadvantages of different methods for seawater salinity measurements.

18.   Table 1-7, please provide reasons why did you only mention the affiliations rather than the name of researchers. In the Tables, most of the researchers are from China, Why dont you cite other researchers from other countries

19.   Table 1-7, please add additional parameters in the Tables, including: refractive index (n), salinity (S), pressure (P), temperature 84 (T), and wavelength (λ)

20.   Point 3.1-3.5, the authors are recommended to provide information about advantages and disadvantages. The authors are found imbalancing the explanations or sometimes did not provide the explanations

21.   Point 4, do you mean accuracy similar to or with sensitivity?

22.   I would like to know the size of sensors used, is that in nano or micrometer? Please provide the info. If possible, add in Table 1-7 the sizes of the sensors

23.   Table 8, what do you mean with difficulties, medium, and easy? It is very unclear, please define it, and it will be best to provide the methods to produce the tools rather than providing quality of the production: medium, easy, difficulties

24.   What do you mean with stability, please provide the standardized word of stability

25.   Table 1-7, please provide how the sensors to be able to test or use in the depth of seawater, how deep the tools can be used? How is about the presssure, and temperatures. Please also refer to my comment point 19

Round 2

Reviewer 1 Report

The authors have revised the manuscript and answered the questions that I was concerned. Thus, I think this paper can be considered for acceptance by Sensors.